# Distinguishing discrete and continuous behavioral variability using warped autoregressive HMMs

**Julia C. Costacurta**
Stanford University
jcostac@stanford.edu

**Lea Duncker**
Stanford University
lduncker@stanford.edu

**Blue Sheffer**
Stanford University

**Winthrop Gillis**
Harvard Medical School

**Caleb Weinreb**
Harvard Medical School

**Jeffrey E. Markowitz**
Georgia Institute of Technology, Emory University

**Sandeep R. Datta**
Harvard Medical School

**Alex H. Williams**
New York University, Flatiron Institute
alex.h.williams@nyu.edu

**Scott W. Linderman**
Stanford University
scott.linderman@stanford.edu

## Abstract

A core goal in systems neuroscience and neuroethology is to understand how neural circuits generate naturalistic behavior. One foundational idea is that complex naturalistic behavior may be composed of sequences of stereotyped behavioral syllables, which combine to generate rich sequences of actions. To investigate this, a common approach is to use autoregressive hidden Markov models (ARHMMs) to segment video into discrete behavioral syllables. While these approaches have been successful in extracting syllables that are interpretable, they fail to account for other forms of behavioral variability, such as differences in speed, which may be better described as continuous in nature. To overcome these limitations, we introduce a class of warped ARHMMs (WARHMM). As is the case in the ARHMM, behavior is modeled as a mixture of autoregressive dynamics. However, the dynamics under each discrete latent state (i.e. each behavioral syllable) are additionally modulated by a continuous latent "warping variable." We present two versions of warped ARHMM in which the warping variable affects the dynamics of each syllable either linearly or nonlinearly. Using depth-camera recordings of freely moving mice, we demonstrate that the failure of ARHMMs to account for continuous behavioral variability results in duplicate cluster assignments. WARHMM achieves similar performance to the standard ARHMM while using fewer behavioral syllables. Further analysis of behavioral measurements in mice demonstrates that WARHMM identifies structure relating to response vigor.

## 1   Introduction

A fundamental question in systems neuroscience is how neural activity generates complex behavior [1–3]. Specifically, a key goal is to understand how changes in neural activity determine which actions are selected or executed on a moment-by-moment basis. To make progress towards this goal, it is essential to study ethologically relevant, naturalistic behavior. A common way of studying naturalistic behavior is to observe animals as they freely explore an environment [4–6]. Such unconstrained,

36th Conference on Neural Information Processing Systems (NeurIPS 2022).

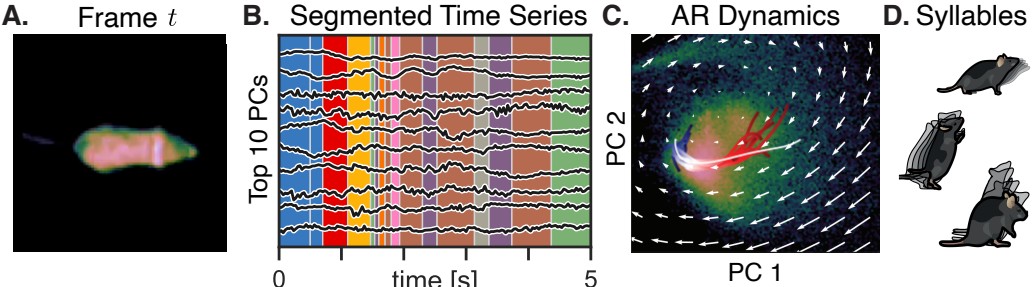

**A.** Frame $t$   **B.** Segmented Time Series   **C.** AR Dynamics   **D.** Syllables

Figure 1: **A.** Single frame of depth video of a freely behaving mouse. **B.** The frames are projected onto their top ten principal components (black lines) and then segmented into discrete syllables (colors) with an autoregressive HMM (ARHMM). **C.** Each syllable is defined by autoregressive (AR) dynamics, which can be visualized as a flow field in PCA space. Here, the scatter plot shows the distribution of frames projected onto the top two PCs, and the red-to-blue trajectories correspond to multiple instances of one discrete state. **D.** Empirically, syllables correspond to interpretable behaviors like investigating, rearing, and falling from a rear, as illustrated here.

spontaneous behavior offers a rich setting for studying behavioral variability. However, the hours of raw video data required to capture naturalistic behavior in detail are high-dimensional and difficult to use for follow-up analyses. Thus, a key goal in behavioral neuroscience research is to develop data-analysis strategies that can extract interpretable lower-dimensional summaries of behavior [4, 5, 7, 7–10]. Ultimately, such descriptions will facilitate relating complex naturalistic behaviors to the underlying neural activity patterns that generate them [3, 11]. This will provide insight into how and why behavior may differ across environmental contexts [2, 3, 11, 12], under pharmacological manipulations [6] or across health and disease [13, 14].

One hypothesis is that the brain generates complex behaviors by concatenating a series of simpler, stereotyped actions [4, 5, 11, 15, 16]. Just like syllables form the building blocks of spoken language, behavioral syllables may be composed to perform complex sequences of behavior. A large focus of previous work has been to discover such behavioral syllables in an unsupervised manner, thereby obtaining a low-dimensional description of high-dimensional behavior [4, 5, 12, 16–19].

Autoregressive Hidden Markov Models (ARHMMs) are well-suited to this task [5, 6]. For example, Wiltschko et al. [5] used depth video to capture the posture of freely moving mice (Fig. 1A). In this work, authors projected the video frames onto the top $D$ principal components (PCs) to obtain a $D$-dimensional time series of behavior. The authors then used an ARHMM to segment the behavioral time series into discrete syllables (Fig. 1B). Each discrete syllable corresponds to vector autoregressive dynamics in PC space, which can be conceptualized as a vector field (Fig. 1C). Each instance of a syllable corresponds to a short, stereotyped trajectory following this vector field (red-to-blue trajectories in Fig. 1C). Empirically, these often correspond to stereotyped patterns of movement like rearing, darting, or grooming (Fig. 1D).

However, current modeling approaches have focused on clustering *discrete* behavioral syllables while failing to account for other, *continuous* forms of behavioral variability. For example, the same type of behavior (e.g. a dart) could be performed more or less vigorously. While these two actions might be identified by a human observer as the same behavior under different speeds (fast vs. slow darts), current models lack the ability to allow for such structured continuous variability within states, and thus might assign these actions to entirely separate behavioral states. As a result, current approaches often *over-segment* video data, by allocating distinct clusters to the same movement type. The data might be better described by a model which incorporates a continuous spectrum of structured variability within a specific behavior.

We extend the ARHMM by incorporating a latent *warping variable*. With this Warped Autoregressive Hidden Markov Model (WARHMM), we are able to capture continuous variability within a discrete syllable. Thus, we are able to disentangle variability due to (discrete) movement type from other forms of (continuous) variability, such as movement speed. We consider two types of warping: *time-warping*, which captures changes in the speed of evolution of the autoregressive dynamics, and *Gaussian process warping*, which allows for nonlinear changes in dynamics. We develop an

efficient inference algorithm for the WARHMM and show that it can correctly identify the underlying states and parameters in simulated data. Then, using behavioral measurements of freely behaving mice, we demonstrate that accounting for continuous sources of behavioral variability can resolve issues of over-segmentation commonly observed using previous approaches. Furthermore, using behavioral data of mice treated with either saline solution or amphetamine, we demonstrate that the WARHMM identifies differences in the distribution of latent warping variables across both groups of mice. This result reflects potential modulations of movement vigor and speed distributions due to the pharmacological intervention.

## 2   Background

We first review related work and then present a brief description of two classes of Hidden Markov Models that have been used for time series segmentation and behavioral modeling. The key features of each model form the basis for our warped extension to the ARHMM model, which represents the main contribution of our work.

**Related work in unsupervised behavioral segmentation.**   We build upon a rich body of work in the unsupervised behavioral segmentation space. Our model is inspired by the ARHMM approach to unsupervised behavioral segmentation (MoSeq) proposed by Wiltschko et al. [5], which is described in the introduction. Berman et al. [4] addressed a similar task in fruit fly video using MotionMapper, which identifies discrete behavioral states as peaks in a non-linear, two-dimensional embedding of postural spectrograms. Hsu and Yttri [17] have proposed B-SOiD, which uses a random forest to classify non-linear postural feature embedding clusters into multiple behavioral classes. Harris et al. [18] fit an autoregressive linear model to time-windowed postural features using low-rank tensor decomposition, and interpret clusters in the fit model parameters as discrete behavioral states. Luxem et al. [19] identify behavioral states by clustering the latent vectors produced by training a variational autoencoder on input from markerless pose estimation [9]. For a more thorough treatment of work in unsupervised behavioral quantification, see McCullough and Goodhill [20] for a recent review. The majority of these related approaches have focused on segmenting behavioral video based on movement type or on tracking animal pose. The model we introduce in section 3 extends on this work by explicitly taking other forms of behavioral variability, such as movement speed, into account.

**Autoregressive Hidden Markov Models.**   An ARHMM (Fig. 2A, top) consists of a discrete latent state variable $z_t \in \{1, 2, ..., K\}$ and observations $\mathbf{x}_t \in \mathbb{R}^D$. Transitions between the discrete state values over time are governed by a transition matrix $\mathbf{P}$, where the entry $\mathbf{P}_{k,k'}$ indicates the probability of advancing from state $z_t = k$ to state $z_{t+1} = k'$. Given the discrete state value, the observations are modeled to evolve according to linear dynamics, where a dynamics matrix $\mathbf{A}_{z_t}$ and bias term $\mathbf{b}_{z_t}$ determine the mapping from $\mathbf{x}_t$ to $\mathbf{x}_{t+1}$ in the presence of Gaussian noise with covariance matrix $\mathbf{Q}_{z_t}$. Thus, the model represents an extension of classic Gaussian mixture models to time series with autoregressive dynamics. The ARHMM model can be summarized as

$$
\begin{aligned}
z_{t+1} \mid z_t = k &\sim \mathrm{Cat}(\mathbf{P}_{k,:}) & k &\in \{1, \ldots, K\} \\
\mathbf{x}_{t+1} \mid \mathbf{x}_t, z_t = k &\sim \mathcal{N}(\mathbf{x}_t + \mathbf{A}_k \mathbf{x}_t + \mathbf{b}_k, \mathbf{Q}_k) & \mathbf{x}_t &\in \mathbb{R}^D
\end{aligned}
\tag{1}
$$

In the context of behavioral modeling [5], the discrete latent state reflects the behavioral syllable (such as a dart, leftward turn, rear, etc.) and determines which dynamics are used to describe the temporal evolution of the observed posture $\mathbf{x}_t$.

**Factorial Hidden Markov Models.**   In Factorial Hidden Markov Models (FHMMs) [21], multiple discrete hidden state variables influence the distribution of observed states. The FHMM model can be summarized as

$$
\begin{aligned}
z_{t+1}^{(i)} \mid z_t^{(i)} = k &\sim \mathrm{Cat}(\mathbf{P}_{k,:}^{(i)}) & k &\in \{1, \ldots, K^{(i)}\}, \ i = 1, \ldots, M \\
\mathbf{x}_t \mid \{z_t^{(i)}\}_{i=1}^M &\sim p(\mathbf{x}_t \mid \{z_t^{(i)}\}_{i=1}^M) & \mathbf{x}_t &\in \mathbb{R}^D
\end{aligned}
\tag{2}
$$

FHMMs are useful in cases when aspects of data can be distributed across multiple states. For example, to fit data that varies according to $M$ binary variables, a standard HMM would need $2^M$ discrete states, while a FHMM would only need $M$ binary states [21].

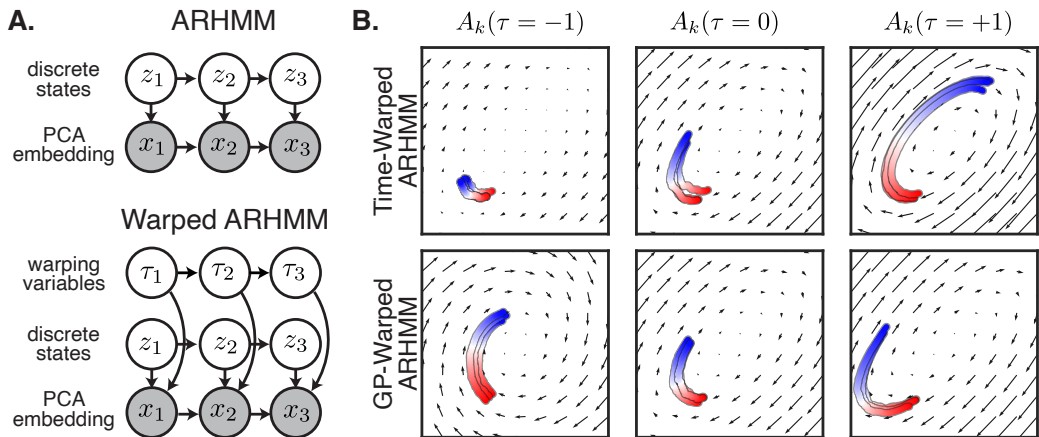

Figure 2: **A.** Probabilistic graphical models for the ARHMM and the Warped ARHMM. The warping variables and discrete states (i.e. syllables) together determine the AR dynamics. **B.** We consider two types of warping: time-warping, where dynamics are sped up or slowed down by the warping variable, and GP warping, where the dynamics vary smoothly but nonlinearly with the warping variable.

In the context of behavior, adding additional latent variables is an appealing way to provide structured variability within discrete syllables. Assuming this variability is included in a reasonable way, an FHMM has the potential to describe a dataset with fewer discrete latent states than an HMM.

## 3   Warped Autoregressive Hidden Markov Models

ARHMMs have been successful in clustering video measurements of behavior into discrete sets of behavioral syllables. However, in practice they are prone to *over-segmentation*, where behaviors that appear similar to a human expert are split into distinct clusters. We hypothesize that oversegmentation arises because the ARHMM conflates discrete sources of behavioral variability — the expression of distinct behavioral syllables — with continuous sources of variability that cannot be captured by linear autoregressive dynamics with Gaussian noise.

To address this limitation, we develop Warped Autoregressive Hidden Markov Models (WARHMMs). WARHMMs extend the ARHMM with an additional latent variable, like in a Factorial HMM. In addition to the discrete state variable $z_t$, WARHMM includes a latent warping state $\tau_t \in [-1, 1]$ at each time step (Fig. 2A, bottom). The warping state $\tau_t$ modulates the dynamics associated with each state $z_t$. While $z_t$ can model rapid switches in dynamics, $\tau_t$ can account for additional variability in the dynamics associated with a given latent state.

The general form of this model class can be summarized as follows, where $\mathbf{A}_k(\tau_t)$, $\mathbf{b}_k(\tau_t)$, and $\mathbf{Q}_k(\tau_t)$ are functions of $\tau_t$:

$$
\begin{aligned}
z_{t+1} \mid z_t = k &\sim \mathrm{Cat}(\mathbf{P}_{k,:}) & k &\in \{1, \ldots, K\} \\
\tau_{t+1} \mid \tau_t &\sim p(\tau_{t+1}|\tau_t) & \tau &\in [-1, 1] \\
\mathbf{x}_{t+1} \mid \mathbf{x}_t, z_t = k, \tau_t &\sim \mathcal{N}\big(\mathbf{x}_t + \mathbf{A}_k(\tau_t)\,\mathbf{x}_t + \mathbf{b}_k(\tau_t), \mathbf{Q}_k(\tau_t)\big) & \mathbf{x}_t &\in \mathbb{R}^D
\end{aligned}
\tag{3}
$$

Below, we consider two possibilities for how exactly the warping variable modulates the dynamics.

### 3.1   Time-Warped ARHMM: linear modulation of autoregressive dynamics

Our first specific instance of WARHMM has a direct motivation in terms of time-warping, and we thus refer to it as a time-warped ARHMM (T-WARHMM). Specifically, we aim to capture continuous changes in how quickly trajectories move through observation space. If we consider the change in current state $\Delta\mathbf{x}_t = \mathbf{x}_{t+1} - \mathbf{x}_t$ due to the dynamics within a given state $z_t = k$ and let $\tau$ be a *log step-size parameter*, we can write

$$
C^{-\tau}\Delta\mathbf{x}_t = \mathbf{A}_k\mathbf{x}_t + \mathbf{b}_k + \mathbf{Q}^{\frac{1}{2}}\epsilon_t, \qquad \epsilon_t \sim \mathcal{N}(0, I)
\tag{4}
$$

$$
\mathbf{x}_{t+1} = \mathbf{x}_t + C^{\tau}(\mathbf{A}_k\mathbf{x}_t + \mathbf{b}_k) + C^{\tau}\mathbf{Q}^{\frac{1}{2}}\epsilon_t
\tag{5}
$$

When $\tau = 0$ the step size $C^\tau$ is always one, so equation (5) is equivalent to the classic ARHMM. For nonzero $\tau$, however, the warping variable modulates how far a single update can move the state, akin to a time-constant in ordinary differential equations. The constant $C$ determines the maximum multiplicative factor by which dynamics can be scaled; in our experiments we set $C = 2$. In terms of the functional mapping, T-WARHMM corresponds to the following mapping between the warping state $\tau_t$ and the parameters determining the autoregressive dynamics of the observed states $\mathbf{x}_t$:

$$\mathbf{A}_{z_t}(\tau_t) = C^{\tau_t}\mathbf{A}_{z_t}, \quad \mathbf{b}_{z_t}(\tau_t) = C^{\tau_t}\mathbf{b}_{z_t}, \quad \mathbf{Q}_{z_t}(\tau) = C^{2\tau_t}\mathbf{Q}_{z_t} \tag{6}$$

Fig. 2B (top) shows how an example syllable's dynamics are sped up or slowed down by changing $\tau$. As $\tau$ increases, the trajectories traverse a greater distance in the same number of time steps. In the context of behavioral modeling, the time-warping variable may capture modulations in movement speed. As we will see in section 5, the inferred warping variables correlate with other intuitive notions of vigor, like centroid velocity in darting syllables, while also offering a quantification of vigor in stationary syllables such as grooming.

## 3.2 Gaussian Process-WARHMM: Nonlinear modulation of autoregressive dynamics

The model formulation for T-WARHMM is interpetable and intuitive, but also makes strong parametric assumptions about how continuous variability affects the dynamics of behavior. As a point of comparison, we consider a more flexible model, in which the influence of $\tau_t$ via $\mathbf{A}_{z_t}(\tau_t)$ and $\mathbf{b}_{z_t}(\tau_t)$ is specified nonparametrically in terms of Gaussian processes. This Gaussian Process WARHMM (GP-WARHMM) moves beyond using $\tau_t$ to describe speed or vigor, as $\tau_t$ could have effects beyond modulation of time-constants of the dynamics. Here $\tau_t$ can modulate entries of the dynamics matrix according to a smooth nonparametric function. In particular, the dynamics are modeled as follows:

$$\begin{aligned} A_{kij}(\tau) &\sim \mathrm{GP}(0, \mathrm{K}_\theta(\tau, \tau')), \quad i, j = 1, \dots, D; \ k = 1, \dots, K \\ b_{ki}(\tau) &\sim \mathrm{GP}(0, \mathrm{K}_\theta(\tau, \tau')), \quad j = 1, \dots, D; \ k = 1, \dots, K \\ \mathbf{Q}_k(\tau) &= \mathbf{Q}_k, \quad\quad\quad\quad\quad\quad\ k = 1, \dots, K. \end{aligned} \tag{7}$$

where $A_{kij}(\tau)$ denotes the $(i, j)$-th entry of the matrix $\mathbf{A}_k(\tau)$ and $b_{ki}(\tau)$ is the $i$-th entry of the vector $\mathbf{b}_k(\tau)$, each modeled as a Gaussian Process with mean zero and covariance function $\mathrm{K}_\theta(\tau, \tau')$. Thus, each coordinate varies smoothly and independently as a function of $\tau$, *a priori*. The prior mean $A_{kij} = b_{ki} = 0$ corresponds to a pause with the state $\mathbf{x}_t$ remaining stationary in expectation. In our experiments, we choose a squared exponential kernel with kernel hyperparameters $\theta = (\rho, \sigma)$.

$$\mathrm{K}_\theta(\tau, \tau') = \rho^2 \exp\left\{ -\frac{1}{2\sigma^2}(\tau - \tau')^2 \right\}, \tag{8}$$

Fig. 2B (bottom) shows how an example syllable's dynamics are modulated by changing $\tau$ in a GP-WARHMM. The central dynamics ($A_k(\tau = 0)$) are the same as in the T-WARHMM above, but now $\tau$ can have nonlinear effects on the dynamics that do more than simply slow them down or speed them up. This extra flexibility could be useful for capturing more general types of continuous variability within discrete syllables, but it may also come at the cost of interpretability.

## 3.3 Inference and Learning

Though we have presented $\tau_t \in [-1, 1]$ as a continuous latent variable, in practice we find it sufficient to discretize $\tau_t$ over a fine grid of $J$ evenly spaced points, since it is a one-dimensional, bounded random variable. Once discretized, we model the warping variable dynamics as a Markov process with a banded transition matrix to encourage small, local changes over time. Similarly, the GP prior in eq. (7) reduces to a multivariate normal after discretization. Full details are provided in the Supplementary Material.

As in standard ARHMMs, we estimate the states and parameters with the Expectation-Maximization algorithm (EM) [22]. We perform exact inference over the discrete syllables and warping variables using the forward-backward algorithm, which runs in $O(T(K^2 + J^2))$ time and uses $O(TKJ)$ memory. For both the time-warped and GP-warped ARHMMs, the parameter updates have closed-form solutions. For large datasets, we used stochastic EM [23] to speed convergence. For the GP-WARHMM, we learned the kernel hyperparameters through optimization of the variational lower bound. Again, complete details are in the Supplementary Material.

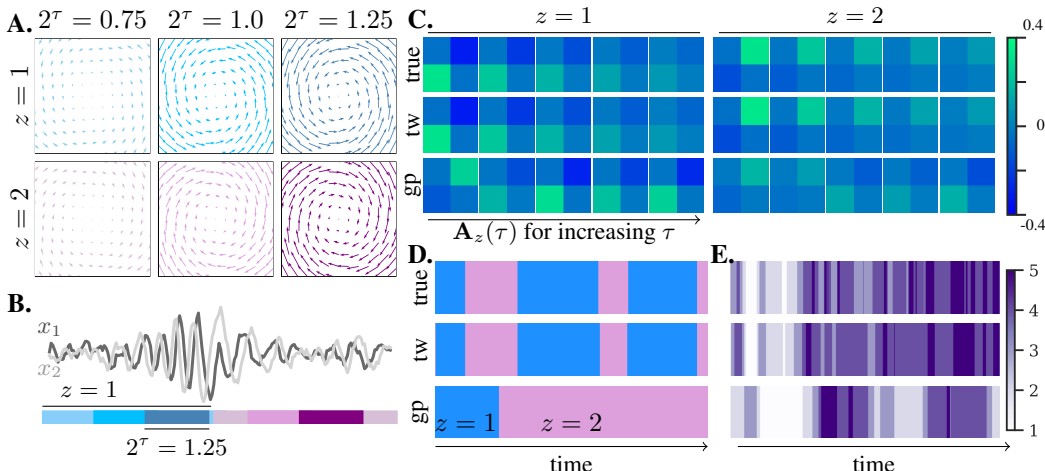

Figure 3: **A.** Illustration of the dynamics of the simulated example. Two discrete latent states represent a clockwise ($z = 2$, bottom) and counterclockwise ($z = 1$, top) rotation, each shown as vector fields. The speed of the rotation is modulated via different values of $\tau$, where $\mathbf{A}_z(\tau) = 2^\tau \mathbf{A}_z$. **B.** Illustration of one instance of the 2-dimensional state trajectory $(x_1(t), x_2(t))$ as the generative dynamics are modulated by both state switches in $z$ (discrete variability, blue versus pink) and changes in rotation angle via $\tau$ (continuous variability, shading of color bar). **C.** The generative values for $2 \times 2$ matrices $\mathbf{A}_z(\tau) = 2^\tau \mathbf{A}_z$ (top row) together with the learned matrices for T-WARHMM (tw, middle) and GP-WARHMM (gp, bottom). T-WARHMM is able to learn the variations in the autoregressive dynamics. GP-TWARHMM learns the overall rotation structure of this example, but the increased flexibility of the GP model makes it harder for it to disentangle discrete and continuous contributions to changes in $\mathbf{A}_k(\tau)$. **D.-E.** The true and inferred latent paths (posterior mode) for $z_t$ (**D**) and $\tau_t$ (**E**) under each model.

## 4 Synthetic data validation

We begin by generating two-dimensional synthetic data $\mathbf{x}_t \in \mathbb{R}^2$ with $K = 2$ discrete states. The autoregressive dynamics under each discrete state are chosen to be a clockwise and counterclockwise rotation. We modulated the dynamics with a time-warped ARHMM, setting $\mathbf{A}_k(\tau) = 2^\tau \mathbf{A}_k$, as illustrated in Fig. 3A. For this simple experiment, we limited the true time-warping variables to $J = 5$ values of $\tau$ evenly spaced on $[-1, 1]$. A snippet of simulated data and the corresponding time constants and states is shown in Fig. 3B. We fit a GP-WARHMM and T-WARHMM with $K = 2$ states and $J = 5$ warping variables to data simulated from this model, and additionally compare results in terms of log-likelihood performance to a classic ARHMM with $K = 2, 4, 10$ discrete states.

| Model | Train LL | Test LL |
|-------|----------|---------|
| T-WARHMM | 0.035 | 0.066 |
| GP-WARHMM | -0.018 | 0.024 |
| 2-state ARHMM | -0.192 | -0.155 |
| 4-state ARHMM | -0.082 | -0.030 |
| 10-state ARHMM | 0.007 | 0.037 |
| True Model | 0.048 | 0.077 |

Table 1: Comparison of test log likelihoods on simulated data across true and fitted models. Higher test LL overall is due to random sampling of train/test data.

The results from this experiment are summarized in Fig. 3C-E and Table 1. Panel C shows the true and learned values of entries of the autoregressive dynamics matrix $\mathbf{A}_z(\tau)$ as both $z$ and $\tau$ vary. We see that both the T-WARHMM and GP-WARHMM learn dynamics that reflect the overall rotational structure of the matrix. However, only T-WARHMM is able to recover the clockwise and counterclockwise variation with $z$ and correct speed modulation with $\tau$.

The increased flexibility of GP-WARHMM, allows it to achieve a test log likelihood which outperforms the $K = 2, 4$ state ARHMMs (Table 1), without correctly disentangling discrete and continuous contributions to the observed data variability. Fig. 3D and E show the true state changes in $z_t$ and $\tau_t$, together with the mode of the inferred posterior distributions under each model. T-WARHMM is able to recover the correct segmentation of the data, while GP-WARHMM's increased flexibility allows it to fit the data, but in a less interpretable way.

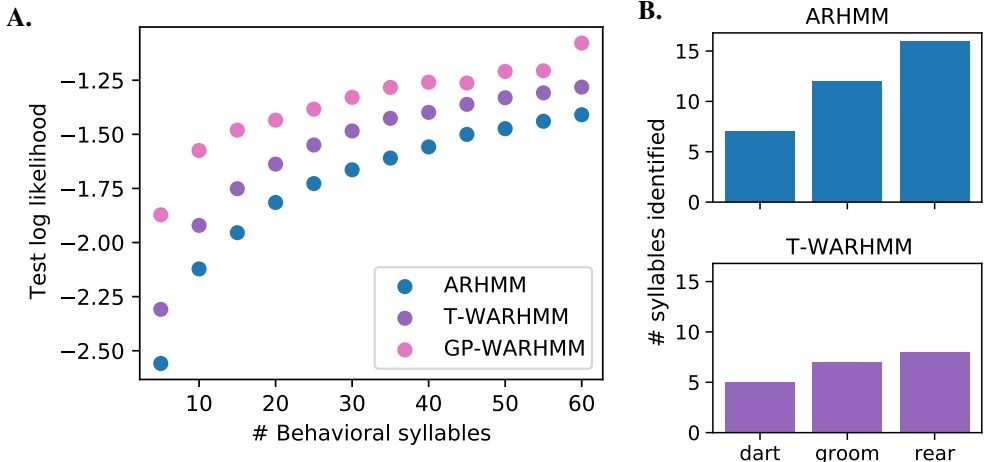

Figure 4: **A.** Comparison of log-likelihood on held-out test data as a function of the number of discrete latent states after training the different model classes on the MoSeq dataset. **B.** For an ARHMM and T-WARHMM with similar test log-likelihoods, total number of syllables that are grouped into the same behavior class by expert neuroethologists. The ARHMM creates new syllables to model qualitatively similar behaviors.

## 5 Modeling depth video of freely behaving mice

For the remainder of the paper, we are interested in extracting discrete and continuous structure in behavior from posture data $\mathbf{x}_t$ extracted from depth-imaging recordings of freely behaving mice. To do this, we reanalyze data from Wiltschko et al. [5], which represents the original application of the ARHMM to clustering mouse behavior. In the context of this dataset, the ARHMM approach is often also referred to as *MoSeq*, and we thus refer to the dataset as the MoSeq dataset.

In the MoSeq dataset, the observations $\mathbf{x}_t$ are taken to be the first 10 principal components of depth camera video data of mice exploring an open field. The dataset consists of 20-minute depth camera recordings of 24 mice. In preprocessing, the videos are cropped and centered around the mouse centroid, and then filtered to remove recording artifacts. Finally, the preprocessed video is projected onto the top principal components to obtain a 10-dimensional time series.

### 5.1 Comparison of model performances on the MoSeq dataset

To provide a direct comparison of performance between the ARHMM, T-WARHMM, and GP-WARHMM, we trained each model using 50 epochs of stochastic EM. The ARHMM is equivalent to setting the number of $\tau$-values in T-WARHMM to $J = 1$. T-WARHMM and GP-WARHMM each had $J = 31$ evenly spaced values of $\tau$ on the interval $[-1, 1]$.

Fig. 4A shows the log-likelihood of each model on held-out test data for a range of $K$ values. For a given value of $K$, both T-WARHMM and GP-WARHMM outperform the classic ARHMM in terms of their generalization performance on unseen data. However, we achieve similar test log-likelihoods for an ARHMM with $K = 40$ syllables, a T-WARHMM with $K = 25$ syllables, and a GP-WARHMM with $K = 10$ syllables. This further illustrates that the ARHMM has to account for continuous variability by creating a larger number of discrete states, while the factorial structure of the WARHMMs enables syllables to be merged when the difference in their dynamics can be explained by changes in $\tau$. The GP-WARHMM outperforms the T-WARHMM in terms of test log-likelihood, which can be attributed to the fact that the GP-WARHMM is more flexible than the T-WARHMM and can capture additional structure in the data. However, the GP-WARHMM accomplishes this at the cost of interpretability. The T-WARHMM warping constant is physiologically motivated and, as we will show in the later results, has direct correlations to centroid velocity and other notions of vigor. While the GP-WARHMM warping variable is allowed to vary the syllable in whichever way will increase data log-likelihood, the T-WARHMM is restricted to modulating syllables in a way that is biologically interpretable. We believe that this interpretability of T-WARHMM makes it more useful for behavioral analysis purposes and thus focus on the utility of T-WARHMM in the remainder.

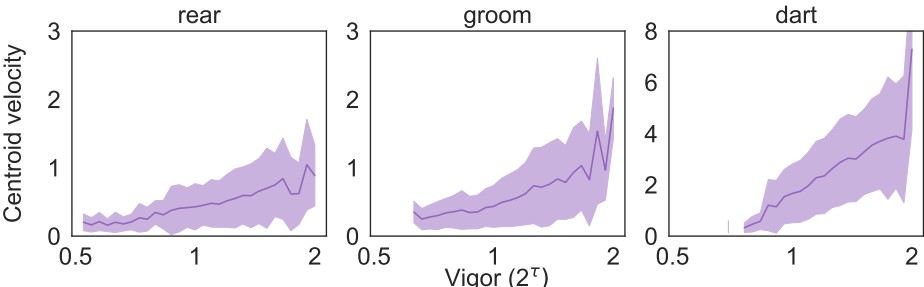

Figure 5: Centroid velocity (pixels/ms) vs. inferred vigor ($2^\tau$) for three representative states of the TW-ARHMM, showing a clear positive correlation between the variables.

## 5.2 Interpreting the time-warping variable

**Which syllables are "warped"?** To begin our analysis of the T-WARHMM results, we fit a model with $K = 20$ discrete states and $J = 31$ time constants. As shown in Fig. 4A, this model attains similar test log-likelihood to an ARHMM with $K = 35$ discrete states. We then determine which types of syllables are over-represented as multiple distinct discrete states in the ARHMM by analyzing the syllable labels from each model. After generating sample videos of behavior from both models, we asked expert neuroethologists and MoSeq users to label the syllables produced. We then gathered the labels under three general behaviors: darts, grooms/pauses, and rears. The results are shown in Fig. 4B. Both models have similar numbers of darting states, while the numbers of grooms and rears are reduced in T-WARHMM, with T-WARHMM halving the number of rear states. While T-WARHMM does not completely resolve the oversegmentation issue, it is able to perform as well as the ARHMM with fewer syllables. More details and video examples of behaviors identified by T-WARHMM are provided in the Supplementary Material.

**Connections to centroid velocity.** From our formulation of the time warping variable, it is expected that the time warping variable and centroid velocity of the mouse would be highly correlated. In Fig. 5 we plot measured centroid velocity (in pixels/ms) vs. inferred vigor ($2^\tau$) for four representative states. We see a strong relationship between centroid velocity and vigor for the darting state, where we would expect such a relationship to occur. The other states also show this relationship, but less consistently. In these states, the warping variable may be accounting for additional forms of timing-related variability that are not well accounted for by centroid speed. These analyses validate that the time warping parameter $\tau$ is able to extract speed-related information also contained in centroid velocity for behavioral syllables such as darting motions.

## 5.3 Identifying drug-induced changes in behavior

Finally, we analyze video recordings of two groups of mice treated with either saline solution or amphetamine. As shown in Fig. 6A, the distribution of centroid speed varies slightly between the groups, with the amphetamine-treated mice performing actions at higher speeds more frequently than the saline mice. We fit a T-WARHMM with $J = 31$ time constants and $K = 20$ syllables to both sets of data. Fig. 6B shows the the inferred vigor ($2^\tau$) distributions from two representative syllables, demonstrating that there is a clear rightward shift between amphetamine and saline treated mice. This indicates that amphetamine mice use faster time warping variables more frequently than saline mice. The difference between means of the $\tau$ index distributions are shown in Fig. 6C. Our model shows a significant difference ($p < 0.05$, independent t-test) between the means of the $\tau$ distributions for all except one of the inferred syllables. Thus, T-WARHMM is able to detect and dynamically track drug-induced differences in behavior across both groups of mice.

## 6 Conclusion

We have introduced an approach for unsupervised behavioral modeling which allows for the identification of continuous variability within discrete behavioral syllables. Our work builds on a prior state-of-the-art technique, which uses a discrete hidden state in combination with autoregressive linear dynamics to cluster behavior. In our proposed model extension (WARHMM), underlying the

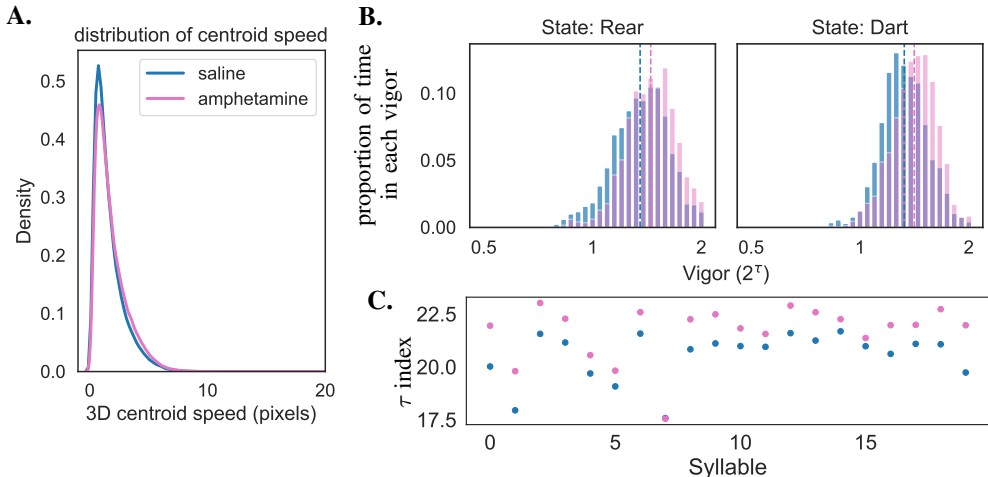

Figure 6: **A.** The distribution of centroid speed across mice treated with saline (blue) or amphetamines (pink). **B.** Histograms showing the vigor distributions across the two classes of mice treated with saline (blue) or amphetamines (pink) for two example syllables (discrete states). The dashed lines indicate the mean of distributions, showing a clear increase in mean vigor in the amphetamine treated group. **C.** Mean $\tau$ index for each of the 20 discrete states in the model for both the saline (blue) and amphetamine (pink) treated mice. The inferred $\tau$ indices in the amphetamine treated group are larger (higher vigor, faster movement speed) for all but one syllable.

behavioral measurement at each time point are both a discrete state variable and a warping variable. Both of these latent variables modulate the dynamics of the syllable, either through discrete switches (via the discrete state variable) or through more continuous forms of modulation (via the warping variable). We found that WARHMM achieved similar performance in terms of log-likelihood on held out test data while utilizing fewer discrete latent states than the ARHMM.

Our analyses on mouse behavioral data focused on an interpretable subclass of warped ARHMM, where the warping variable interacts linearly with the autoregressive dynamics of the observed data (the T-WARHMM). Notably, the warping variable in this simple model can be viewed as implementing a local rescaling of time. Similar time warping models have been fruitfully applied to trial-structured neural data [24, 25], and our work illustrates how these ideas can be extended to naturalistic time series data that lack repeated trial-structure. Our work on mouse behavioral data illustrates that continuous variability in behavior may be induced through pharamacological interventions and can be extracted directly from video recordings.

**Limitations.** For simplicity, we discretized $\tau$ on a fine grid instead of explicitly performing inference over the continuous variable. This was done in favor of computational efficiency and ease of inference. While we believe the fine grid we've used for $\tau$ offers a good approximation of a continuous function, future work could capture continuous variability by including a truly continuous posterior, e.g. via a Gaussian Process [24]. Furthermore, our model assumes that the behavioral PCs evolve according to linear dynamics, and though the GP-WARHMM allows these dynamics to be modulated in nonlinear ways, the basic assumption could be relaxed to more general nonlinear dynamics in future work.

Unsupervised deep learning approaches exist to address similar problems in the behavioral segmentation space, e.g. B-SOiD [17]. Another approach could involve including speed as an input and clustering features for these methods. However, since clusters are defined based on differences in input features, we predict that this approach would lead to further segmentation of syllables. In contrast, the motivation behind WARHMM is to collapse syllables together by allowing for structured variation within a single syllable.

**Future work.** This work establishes multiple directions for future research. The first involves further extending the ARHMM model by adding in additional forms of structured, interpretable variability. What aspects of behavior, in addition to speed, may vary within syllables? In the context of mouse behavior, variables such as turning radius or depth-camera pixel height may provide additional

continuous axes along which behaviors can be modulated. Another future direction lies in using T-WARHMM to analyze neural correlates of behavior. Similar behavioral segmentation approaches have been used to simultaneously analyze the evolution of behavioral and neural recordings [8, 26–28]. A future research area of particular interest lies in understanding how the dorsolateral striatum encodes action selection and vigor [13, 29–32]. Neural activity in this midbrain nucleus is known to be disrupted in motor diseases like Parkinson's and Huntington's [33, 34]. The T-WARHMM approach allows us to simultaneously extract descriptions of movement type (how frequently actions are selected) and movement speed (how vigorously actions are performed) from behavioral measurements. We expect that the ability to disentangle the contributions of these two factors could provide an important basis for better understanding how variations in neural activity determine changes in action selection and vigor. Finally, the models we have proposed here could be applied more widely to analyze other types of behavioral data.

## Acknowledgments and Disclosure of Funding

We would like to thank Andy Warrington, Laura Driscoll, Libby Zhang, Jimmy Smith, Michael Salerno, Mohammed Osman, Sherry Lin, Akshay Jaggi, and our reviewers for their feedback and suggestions.

This work is supported by the Simons Collaboration on the Global Brain (SCGB 697092) and NIH BRAIN Initiative Grant 1U19NS113201-01. JCC is funded by the NSF GRFP and Stanford Graduate Fellowship. SRD is supported by NIH grants U19NS113201, RF1AG073625, R01NS114020, the Brain Research Foundation, and the Simons Collaboration on the Global Brain. JM is supported by a Career Award at the Scientific Interface (CASI) from the Burroughs Wellcome Fund. WG is supported by NIH grant F31NS113385. SWL is additionally supported by the Sloan Foundation.

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
