# OpenReview forum: "Distinguishing discrete and continuous behavioral variability using warped autoregressive HMMs"
_NeurIPS.cc/2022/Conference — NeurIPS 2022 Accept_

### Official Review · Reviewer_4Fyi · 2022-07-09

**Rating:** 7
**Confidence:** 4
**Soundness:** 4 excellent
**Presentation:** 4 excellent
**Contribution:** 3 good

**Summary:**

This is an application + algorithm paper. Application-wise, it focuses on the summarization of videos of freely moving mice for the purpose of neuroscience research. For this purpose, a new HMM model is proposed, whose state space is such that a discrete state variable captures discrete behaviors (sometimes referred to as movement primitives) and a continuous variable captures an element of warping of the dynamics within discrete states.
More precisely, the paper focuses on modeling of continuous-valued multivariate time-series observations (such as a video stream that has been reduced to a time-series of principal components). It starts from Auto Regressive Hidden Markov Model (ARHMM) models, in which within every discrete state z_i, (spanning some variable length of time), the observations within the state are modeled by a linear-gaussian model (i.e. linear dynamics with gaussian noise) whose parameters (A, b and Q) are state dependent. The paper then suggests a novel extension of the ARHMM model in which an additional continuous variable, tau (discretized for sake of computation) is added in 2 possible ways, both of which allow the linear-gaussian model for z_i to be "warped:. In the version given more attention within the paper, the dynamics are simply sped up or down by 2^tau (tau in [-1 - 1]). The other warping method involves a gaussian process which, while fitting realt data better, is discarded from further analysis due to lack of interpretability.
The method is applied to (pre-processed) videos (IR grayscale + depth) of mice "freely exploring" within their lab environment. The paper shows how the proposed Time-Warped-AHRMM (T-WARHMM) can achieve log-likelihood values that are better than ARHMM for the same number of discrete hidden states (or equivalent to an ARHMM with more hidden states). This is explained by the ARHMM requiring several states to represent the same dynamic at different speeds whereas the WARHMM can do so with the addition of the warping hidden variable. The estimated warping variable, can be interpreted as a measure of "rigorousness" and thus conveys biologically-relevant meaning.


**Questions:**

* In equation 7,  the entries of A and b are modeled as GP(0, K(tau, tau`)). Does 0 refer to a zero mean? aren't these equations missing a base A and b which is only modified by a zero-mean GP?

* The results in figure 3 and table 1 (for a synthetic time series) could be extended to include the regular ARHMM (which serves as a baseline in this study. I'd be interested to see how well that model captures the discrete state (perhaps as a function of the number of available model "syllables”), as was done in the "real-data" experiment.


**Limitations:**

I see no  limitations within the context of behavioral analysis nor any potential negative societal impact of this work.

**Strengths And Weaknesses:**

I find the modeling sound and so is the interpretation of the experimental results. The benefits of such modeling for automating the annotation of behavioral videos of mice with the purpose of relating their behavior (as annotated by the model) to experiment conditions is clear. The model and its inference method is relatively clear. So are the experimental results.

The results are limited to a single non-public real dataset. It is therefore hard to assess whether the new method is a good fit to the particular application (analysis of behavior with discrete types of actions that can be warped in time). An analysis of a larger set of datasets (or one which is comprehensive and publicly available) would have been more convincing.

The MoSeq dataset seems to lack ground-truth annotations of the behaviors. I would have liked to see an analysis where a ground-truth annotation can be compared against the HMM so that a measure can be calculated which captures how well the HMM states capture behavior that the researchers care about. For example, I assume that MoSeq or some other dataset exists where (expert) human annotations are available, segmenting the video into human-recognisable behaviors such as those mentioned in the paper (dart, groom, rear and, I presume, "unspecified"). I would then like to see **how well HMM states fit within human annotated states** (as opposed to the results in the paper were states where assigned to behaviors by experts) and whether WARHMMs are better under _that_ sort of performance measure (which would be more relevant to behavioral scientists).

---

> ### Author Response · Authors · 2022-08-01
> **Response to reviewer 4Fyi**
>
> Thank you for your careful reading of our submission and valuable feedback.
> We agree that working with publicly available datasets is important for reproducibility. Please note that the dataset we used in this paper is also available in combination with access to the original MoSeq code for analysis of this dataset (https://dattalab.github.io/moseq2-website/index.html). Likewise, similar datasets are available publicly (see e.g. here https://github.com/dattalab/moseq-drugs) We have updated the manuscript now to highlight that the data is publicly available. We will also make our code publicly available on GitHub upon publication.
>
> Regarding your point about human annotated datasets, we agree that this would be a nice and meaningful inclusion. However, such annotations do not exist for the Moseq dataset, and, similarly, we are not aware of a labeled dataset for subsecond structure in behavioral measurements in general. The labels that were added by our experimental collaborators post-hoc are much coarser than those identified by the model. In reality, e.g. a “rear” is composed of multiple syllables, with individual syllables capturing e.g. the beginning, middle or end. The human expert labeling grouped all of these syllables within one category and thus lacks granularity at the sub-second level. If such a labeled dataset were available, however, we agree that it would be a meaningful inclusion and might also reveal interesting biases across human labels and unsupervised labeling using our method. For now, we verified the interpretability of model-identified clusters through post-hoc analysis (“crowd videos”), examples of which are included in the supplemental material. We note that this fully unsupervised approach is common within the field and has led to many scientific insights [1,2,3,4,5]. In the future, it could be interesting to extend our model to explicitly incorporate labels as semi-supervised learning signals (as has been done in pose tracking [6]).
>
> In equation 7, $GP(0, K(\tau, \tau'))$ does indeed refer to a Gaussian process with a zero-mean function and a covariance function described by  $K(\tau, \tau')$. The state transitions are described as $x_{t+1} = (I - A(\tau)) x_t + b (\tau)$ + noise. Thus, when A and b are both zero the state stays stationary in expectation. We have updated the relevant section in the manuscript to clarify this point.
>
> Lastly, we have included results on synthetic data for the classic AR-HMM with 2, 4, and 10 discrete states and have updated Table 1 to reflect this. We agree that this serves as a valuable reference point for the performance of the WARHMM variants.
>
> [1] Wiltschko, A. B., Johnson, M. J., Iurilli, G., Peterson, R. E., Katon, J. M., Pashkovski, S. L., ... & Datta, S. R. (2015). Mapping sub-second structure in mouse behavior. Neuron, 88(6), 1121-1135.
>
> [2] Markowitz, J. E., Gillis, W. F., Beron, C. C., Neufeld, S. Q., Robertson, K., Bhagat, N. D., ... & Datta, S. R. (2018). The striatum organizes 3D behavior via moment-to-moment action selection. Cell, 174(1), 44-58.
>
> [3] Wiltschko, A. B., Tsukahara, T., Zeine, A., Anyoha, R., Gillis, W. F., Markowitz, J. E., ... & Datta, S. R. (2020). Revealing the structure of pharmacobehavioral space through motion sequencing. Nature neuroscience, 23(11), 1433-1443.
>
> [4] Chen, J., Markowitz, J.E., Lilascharoen, V. et al. (2021). Flexible scaling and persistence of social vocal communication. Nature 593, 108–113.
>
> [5] Akiti, K., Tsutsui-Kimura, I., Xie, Y., Mathis, A., Markowitz, J., Anyoha, R., ... & Watabe-Uchida, M. (2021). Striatal dopamine explains novelty-induced behavioral dynamics and individual variability in threat prediction. BioRxiv.
>
> [6] Blau, A., Gebhardt, C., Bendesky, A., Paninski, L., & Wu, A. (2022). SemiMultiPose: A Semi-supervised Multi-animal Pose Estimation Framework. arXiv preprint arXiv:2204.07072.

---

### Official Review · Reviewer_N8jX · 2022-07-11

**Rating:** 6
**Confidence:** 3
**Soundness:** 3 good
**Presentation:** 3 good
**Contribution:** 2 fair

**Summary:**

This paper provides an extension to autoregressive Hidden Markov Models (ARHMM), in which the state transitions are modulated by a temporal component that controls the rate of state change. The paper provides two variants: in one of them, the step size is a parametric (time-wrapped), and the other one is based on Gaussian Processes. The model is applied to two datasets, one synthetic and one real-world dataset (MoSeq). The results on the synthetic dataset show that the model generally can infer the hidden state and also the temporal dynamic. The results on the real dataset show that the vanilla ARHMM can reach a similar performance (in terms of test likelihood) as the proposed model but requires a higher number of hidden states, which might hurt the interpretability of the model.


**Questions:**

C is chosen to be 2, and \tau is within [-1, 1], which limits how much states can change. Whare the basis for this assumption?


**Strengths And Weaknesses:**

Strengths:
- The motivation for the model is sound, and it addressed an interesting problem with ARHMM.
- The proposed models are simple but motivated by the target problem.
- The presentation of the paper is good, and the paper is easy to follow.

Weaknesses:
- The method is not compared to any state-of-the-art action recognition/segmentation techniques (such as deep-learning approaches).
- The choice of hyper-parameters is rather unclear. It is mentioned in the supplementary section that a range value for \sigma was tried, which didn't affect the performance much. Are these performances on the train or test set? I'd expect the parameters to be tuned on a separate validation set.

---

> ### Author Response · Authors · 2022-08-01
> **Response to reviewer N8jX**
>
> Thank you for your careful reading of our submission and valuable feedback.
>
> We understand why comparisons to e.g. deep learning approaches for behavioral modeling would be interesting. However, we are not aware of any existing approaches that have been applied to extract sub-second discrete structure from behavioral time-series (like the AR-HMM does), while also handling the separation into continuous and discrete forms of variability – the key motivation of our proposed approach. Many current deep learning approaches focus on tracking  [1, 2, 3],  clustering natural behavior on a much coarser time-scale [3], or finding nonlinear embeddings of video to then feed into AR-HMMs for segmentation [4,5]. In addition to this, many of these methods tend to require at least some form of supervision using labeled training data, which is not available in the completely unsupervised setting we consider here. While we do agree that comparisons to state-of-the-art approaches addressing the same problem would be important, we are not aware of any such published work and would welcome any references that the reviewer had in mind.
> Overall, our goal was to address a significant shortcoming of a widely used method [6,7,8,9,10] and provide an interpretable and valuable extension. Thus, while we agree that additional comparisons are valuable for future work, we are not aware of an approach that has already solved this problem and would warrant comparison at this time.
>
> Regarding the hyperparameter tuning for $\sigma$, we have updated the method to automatically learn this parameter value through optimization of the variational lower bound (as part of the M step in EM). We have updated the relevant sections and figures with the new fits in the revised manuscript.
>
> Lastly, regarding the choice of $C$ and $\tau$, we have made the choices to include prior beliefs about the range of speed variability in natural behavior. In particular, we assumed that a single behavioral syllable should encapsulate behaviors performed at approximately half to twice the “base” speed. However, since $C$ determines the range of $\tau$, choosing a large value for $C$ may result in poorer algorithm performance (see step-size comment in response to reviewer w5pw), while choosing a very small value would decrease the amount of variability covered in a single syllable. In addition to this, we note that the discretization is an approximation we make for computational tractability. The work we have presented here could be extended in scope to infer $g=C^\tau$ as a continuous variable (e.g. as a Gaussian process) without the need for explicit a priori choices for C and \tau. We have updated Section B.2 of the supplementary material to clarify this point and to highlight the generality of these choices.
>
> [1] Mathis, A., Mamidanna, P., Cury, K. M., Abe, T., Murthy, V. N., Mathis, M. W., & Bethge, M. (2018). DeepLabCut: markerless pose estimation of user-defined body parts with deep learning. Nature neuroscience, 21(9), 1281-1289.
>
> [2] Pereira, T. D., Aldarondo, D. E., Willmore, L., Kislin, M., Wang, S. S. H., Murthy, M., & Shaevitz, J. W. (2019). Fast animal pose estimation using deep neural networks. Nature methods, 16(1), 117-125.
>
> [3] Segalin, C., Williams, J., Karigo, T., Hui, M., Zelikowsky, M., ... & Kennedy, A. (2021). The Mouse Action Recognition System (MARS) software pipeline for automated analysis of social behaviors in mice. Elife, 10.
>
> [4] Batty, E., Whiteway, M., Saxena, S., Biderman, D., Abe, T., ... & Paninski, L. (2019). BehaveNet: nonlinear embedding and Bayesian neural decoding of behavioral videos. Advances in Neural Information Processing Systems, 32.
>
> [5] Johnson, M. J., Duvenaud, D. K., Wiltschko, A., Adams, R. P., & Datta, S. R. (2016). Composing graphical models with neural networks for structured representations and fast inference. Advances in neural information processing systems, 29.
>
> [6] Wiltschko, A. B., Johnson, M. J., Iurilli, G., Peterson, R. E., Katon, J. M., ... & Datta, S. R. (2015). Mapping sub-second structure in mouse behavior. Neuron, 88(6), 1121-1135.
>
> [7] Markowitz, J. E., Gillis, W. F., Beron, C. C., Neufeld, S. Q., Robertson, K., ... & Datta, S. R. (2018). The striatum organizes 3D behavior via moment-to-moment action selection. Cell, 174(1), 44-58.
>
> [8] Wiltschko, A. B., Tsukahara, T., Zeine, A., Anyoha, R., Gillis, W. F., ... & Datta, S. R. (2020). Revealing the structure of pharmacobehavioral space through motion sequencing. Nature neuroscience, 23(11), 1433-1443.
>
> [9] Chen, J., Markowitz, J.E., Lilascharoen, V. et al. (2021). Flexible scaling and persistence of social vocal communication. Nature 593, 108–113.
>
> [10] Akiti, K., Tsutsui-Kimura, I., Xie, Y., Mathis, A., Markowitz, J., ... & Watabe-Uchida, M. (2021). Striatal dopamine explains novelty-induced behavioral dynamics and individual variability in threat prediction. BioRxiv.

---

> > ### Comment · Reviewer_N8jX · 2022-08-05
> > **Baselines**
> >
> > Thank you for your response.
> >
> > With regard to comparison with other baselines, several references are cited in section "Related work in unsupervised behavioral segmentation" and below. I appreciate that these models may not have an explicit mechanism to capture continuous behavioral variabilities (such as vigor), but they may still be able to capture such variabilities through speed-related input features (like B-SOiD and perhaps other works) or implicitly learn about them. From the current results, it is unclear how the model performs in comparison with such previous works, for example, in terms of Fig 4.

---

> > > ### Author Response · Authors · 2022-08-05
> > > **Response to Baselines comment**
> > >
> > > Thank you for your continued engagement in this discussion. We agree that B-SOiD (and related methods for obtaining behavioral representations, like MotionMapper) present interesting alternatives for finding discrete clusters of behavior. As you suggest, one could include speed as an input feature to these methods. However, note that B-SOiD produces clusters based on *differences* in input features. Thus, it would have the same speed-based over-segmentation problem as MoSeq. In contrast, the motivation behind WARHMM is to collapse syllables together by allowing speed to vary *within* a single syllable.
> > >
> > > We would also like to point out that pipelines like B-SOiD are not generative models, i.e. they don’t define a distribution over the data. Thus, we could not compare B-SOiD to the ARHMM-based models using test log-likelihood, as in Fig. 4. We agree that a systematic comparison of these different approaches along other dimensions would be interesting, but we think that would merit a separate investigation from this paper, which is focused on an interpretable extension of the MoSeq generative model.

---

> > > > ### Comment · Reviewer_N8jX · 2022-08-07
> > > > **reply**
> > > >
> > > > "it would have the same speed-based over-segmentation problem as MoSeq." Unfortuantly it is hard to assess this statement without seeing experimental results. Therefore I still think the lack of proper comparison with other baselines that aim to address the same problem is a weakness of this paper.

---

> > > > > ### Author Response · Authors · 2022-08-09
> > > > > **Response to Reviewer N8jX**
> > > > >
> > > > > Thank you for your time, comments, and investment in the improvement of our paper. We will investigate B-SOiD and update the paper accordingly if accepted.
> > > > >
> > > > > We wish to reiterate a few points in the meantime: first, that the main motivation for the paper is to improve upon a well-used method (MoSeq), and we show that WARHMM does add functionality to and improve performance of this standard model. We acknowledge that MoSeq-like methods are not the only way to get useful unsupervised representations of behavior. However, when using methods that cluster input features, we posit that the addition of centroid speed as an input provides an additional dimension along which to *divide* clusters, while the key feature of WARHMM is that it allows for wider variation *within* clusters.

---

### Official Review · Reviewer_w5pw · 2022-07-11

**Rating:** 7
**Confidence:** 4
**Soundness:** 4 excellent
**Presentation:** 3 good
**Contribution:** 3 good

**Summary:**

This paper develops a useful extension to the unsupervised behavioral syllable segmentation model space (specifically ARHMM), with an additional latent variable for time warping (handled as in a factorial HMM fashion). Time warping is achieved by having a log step-size parameter that allows the behavior to speed up or slow down dynamically. The model is validated in synthetic data, the applied to analyze mouse behavior data. The results demonstrate that the time-warping method enables clustering together same behavior type but with different speed/rigor (e.g., fast vs. slow runs), which were often identified as distinct behavioral syllables in previous models. The paper also proposed a more flexible model, GP-warping, but the results indicate that GP warping is less interpretable than the more straightforward time-warping.


**Questions:**

i) Step sizes: With the time-warping model, do you observe/anticipate any overshooting issues when the step sizes are too large? The dynamical matrices shown in the examples are simple and smooth rotations, but in reality the dynamics can have more structure with varying scale and complexity. In practice, would there be any signatures in the results that tell you when the step size is too large for the specific dynamics?

ii) Figure 3: I think I got most of the messages that the authors wanted to convey here, but it required going back and forth multiple times between text and captions and some educated guess. Some color legends and labels would be useful. Some things I still have not understood: Why are there four colors in each box in Fig 3C, and what are these colors? Also, what are x1 and x2 in Fig 3B; are they just two realizations from the same latent states or is something different? With improved figure annotations, my clarity rating for this paper could go up to 4.

iii) Description of GP model results on synthetic data: I am generally happy with how clearly and transparently the paper is written, but I feel that the GP-warped model results in the synthetic data section is a little sugar coated... For example, L195: "... GP-WARHMM's increased flexibility allows it to fit the data, but in a less interpretable way". From Fig 3 and Table 1, I would say that the GP model is simply not fitting the data very well, at least as it is. (Please feel free to correct me though; I might have mis-read Fig 3.) Of course you don't need to be as blunt, and I understand that it is constructive to put things on a positive context. But as I mentioned in the strengths/weaknesses above, the consideration of GP model can be valuable even if the results are not as good, and I think the authors can be more straightforward.


**Limitations:**

The authors have addressed the limitations of their methods and assumptions. They also discussed the potential societal impact of their work in the supplementary material.


**Strengths And Weaknesses:**

This is a nice paper and certainly an addition to the community assets. The presented method will be readily useful for more interpretable analysis of natural behavior. The paper followed good practice: well defined problems, reasonable choice of models, validation on synthetic data, and finally application to real data. The paper is very clear and well written.

I also appreciate that the paper considered two models (time-warping and GP-warping) as examples of interpretable vs. flexible models, and was clear about their assumptions and performances. Although the GP-warping model performance was not as good on synthetic data, which might count as a weakness per se, I think it was still valuable to share this part of the research and would like to credit the authors for doing so. The juxtaposition of two models provided a better context to the choice of time-warping model.

A weakness of the paper might be that, from the technical viewpoint, the method is a relatively small extension of the existing models. However, it is an extension that brings value both practically (more interpretable data analysis with less syllables) and conceptually (allowing the timescale variability can be important in behavior analysis).

---

> ### Author Response · Authors · 2022-08-01
> **Response to reviewer w5pw**
>
> Thank you for your careful reading of our submission, valuable feedback and for agreeing that our work represents a valuable contribution.
>
> Regarding your point about overshooting and step-sizes in the time warping model: Since we model dynamics as conditionally linear, the complexity within each syllable is fairly limited. If the step-sizes are too large (due to a larger inferred value of the time-warping variable, or choosing a large value for the $C$ hyperparameter) we would expect to either see instabilities in the dynamical system, or dynamics which are effectively pushed towards the identity (corresponding to a pause, behaviorally). Since the time-warping is inferred from data, we would only expect this to happen when data is too noisy or limited to fit the model. In this case, the model could trade off a large step-size with multiple switches in the discrete latent state variable to effectively “mask” any resulting instabilities in any individual dynamics. We have not encountered this in practice, however. We’ve included a discussion of the choice of the $C$ hyperparameter in Section B.2 of the supplementary material in the updated submission. In future work, one could estimate the hyperparameter $C$ by maximizing the marginal likelihood of the data. Since $C$ is only a scalar hyperparameter, we chose to manually set it to a biologically plausible limit for the maximum scaling of the dynamics.
>
> Thank you for your comments regarding the clarity of Figure 3. We have expanded on the figure legend to address your points in the revised manuscript. In particular, we have added color legends to Fig. 3C-E, have changed the colormap of Fig. 3C to avoid overlap with Fig. 3A-B+D, and have made the following caption clarifications:
> For Fig. 3B: Illustration of one instance of the 2-dimensional state trajectory (x1(t), x2(t)) as the generative dynamics are modulated by both state switches in z (discrete variability, blue versus pink) and changes in rotation angle via \tau (continuous variability, color shade).
> For Fig. 3C: The generative values for 2 x 2 matrices $ \mathbf{A}_z(\tau) = 2^\tau  \mathbf{A}_z$ (top row) together with the learned matrices for T-WARHMM (tw, middle) and GP-WARHMM (gp, bottom).
>
> Regarding the GP model, we agree that the results in their current form mostly highlight the shortcomings of the GP model and do not demonstrate that the test likelihood performance still reflects a good fit for the data. We have expanded on the synthetic data results by adding the performance of an AR-HMM with two, four, and ten discrete states to better demonstrate that the GP-WARHMM model can fit the observed data well (in terms of the marginal likelihood performance on both training and test data) but fails to infer the underlying latent variable structure of the generative process (in terms of separating continuous and discrete sources of variability) in Fig. 3. We have added both training and test set performance across models to Table 1, and have included additional results using the AR-HMM with two, four and ten discrete states as a reference point. We hope that these additional results better demonstrate our point regarding the GP model being more flexible but less interpretable.

---

> > ### Comment · Reviewer_w5pw · 2022-08-06
> > **response to authors**
> >
> > I would like to thank the authors for providing the rebuttal comment as well as the revised version of the paper.
> >
> > i) The explanation on the step sizes, overshooting and the hyperparameter C makes sense, and it also helps me understand the method better.
> >
> > ii) The new Figure 3 is much more informative! Thank you for making these changes.
> >
> > iii) Regarding the GP model: I appreciate the additional results. Indeed, if I only look at the test set performances in the new Table 1, it does look like the GP model outperforms the ordinary ARHMM model with similar numbers of latent states. That said, I got confused on a potentially basic point: why are the test likelihoods are significantly higher than the train likelihoods?

---

> > > ### Author Response · Authors · 2022-08-06
> > > **Log Likelihoods**
> > >
> > > We're glad that the revisions we've made have clarified these points for you, and appreciate that your comments have made the paper more readily understandable.
> > >
> > > Regarding the test/train likelihoods: this is due to the randomly sampled train and test data we used to train the models. In the simulated data presented in the paper, the test dataset happened to have higher likelihood on average, indicated by the row of likelihoods corresponding with the true model. We have fit the models on a few more synthetic datasets generated from the same true model, and have confirmed that the test likelihoods are not consistently higher than the train likelihoods. We will clarify this in the text to address any confusion that might arise.

---

> > > > ### Comment · Reviewer_w5pw · 2022-08-09
> > > > **response to authors**
> > > >
> > > > Thank you for the clarification about the test/train likelihoods.

---

### Meta-Review · Area_Chair_wndE · 2022-08-20

**Recommendation:** Accept
**Confidence:** Certain

**Metareview:**

This paper introduces warped auto-regressive HMMs, where both discrete state transitions z_{t+1} | z_t and continuous observations x_{t+1} | x_t, z_t, depend on the previous time step, and where the linear dynamics for observation transitions are either time-warped (i.e., the linear transition matrix and bias are multiplied by a time-dependent latent step size) or obtained by Gaussian Process regression with latent prototypes. The method is evaluated and applied to modeling “syllables” of behaviour in free-moving mice in neuroscience experiments directly from depth camera recordings.

Reviewers praised the clarity and structure of the paper (w5pw, N8jX, 4Fyi), the motivation for the paper (N8jX, 4Fyi), the honesty in reporting results on slightly worse GP warped ARHMM (w5pw)

The reviewers noted that the contribution was incremental (w5pw) and did not compare to any modern deep learning baseline (N8jX) - for the latter, the authors partially addressed this point by responding that existing work did not handle tracking and segmentation without supervised training. Reviewers noted that evaluation was limited to a proprietary dataset (it is actually publicly accessible) without human ground-truth annotations (4Fyi) and wished more evaluations had been done - the authors replied that results were reviewed by experimentalists. They also had some questions regarding figures that were properly addressed (w5pw).

Reviewers agree on high scores (6, 7, 7) and therefore I would recommend this paper for acceptance.

Sincerely,
Area Chair

**Award:**

Yes

---

### Decision · Program_Chairs · 2022-09-14

Accept